# What choices of visual correction do people make for recreational running?

**Keziah Latham**[1]*, **Georgia Dunn**[2], **Andrew Morrison**[2], **Matthew A. Timmis**[2]

1 Vision and Hearing Sciences Research Centre, Anglia Ruskin University, Cambridge, Cambridgeshire, United Kingdom, 2 Cambridge Centre for Sport and Exercise Science, Anglia Ruskin University, Cambridge, Cambridgeshire, United Kingdom

☯ These authors contributed equally to this work.
* Keziah.latham@aru.ac.uk

**Data Availability Statement:** All relevant data are within the manuscript and its Supporting Information files, and are available online at doi.org/10.25411/aru.25109270.v1.

## Abstract

### Purpose

Recreational runners who need refractive visual correction will need to choose the type of visual correction (spectacles, contact lenses, or no correction) to wear when running. The prevalence of correction choices and distinguishing demographic features associated with these choices of correction were investigated.

### Methods

A 26-item online questionnaire covering demographics, running habits, and visual corrections used was answered by 941 people. Participants were aged 18 years or older, considered themselves a recreational runner and reported needing refractive correction for everyday life. Bivariate analyses and binary logistic regression were used to determine the variables independently associated with including or avoiding types of correction for running.

### Results & conclusions

Participants ran an average of 3.8times, 3.6hours, and 33.4km per week. Running habits were not associated with type of visual correction, suggesting that visual correction does not present a barrier to participating in recreational running. Solely using spectacles for running was the most common choice (38.8%) and they were worn for running at least sometimes by 65.8%. Fewer people wore spectacles for running than for general purposes (98.9%), suggesting spectacles have limitations for running. Spectacle wear for running was significantly associated with not having contact lenses available and longer duration of wearing visual correction. Running without correction was a common choice, with 15.5% of the sample only running without correction, and 26.2% running at least sometimes without their visual correction. Not wearing correction was significantly associated with not having contact lenses, a shorter duration of wearing visual correction, and a lower myopic refractive error. Contact lenses were the sole correction for running for 15.6% and were worn for running at least sometimes by 40.5%. Contact lenses were significantly more likely to be used

**Funding:** The author(s) received no specific funding for this work.

**Competing interests:** The authors have declared that no competing interests exist.

by women and younger runners. Signposting runners to contact lenses if this has not previously been considered is recommended.

## Introduction

Refractive error is a common visual issue. It is estimated that 27% of European adults are myopic (short-sighted), 23% are hyperopic (long-sighted), and 40% have astigmatism (imperfection in the curvature of the eye) [1]. Further, 39% of the population of Western Europe are presbyopic (reduced ability to focus on nearby objects) [2], requiring different refractive corrections for distance and near tasks.

In the general population, refractive error is most commonly corrected using spectacles. Whilst using spectacles as a form of distance vision correction for daily living tasks (e.g., shopping and driving) may place little impact on quality of life beyond the wearer's opinion regarding their appearance [3], challenges with wearing spectacles may arise when participating in moderate-high levels of physical activity.

As a recent example, Somerset and England Cricketer Jack Leach received media attention for repeatedly cleaning his glasses while batting against Australia in the third Ashes test in 2019 [4]:

> "*I was just trying to do everything I could to see the ball as well as I could. I was sweating and I've obviously got no hair so the sweat's dropping into my glasses and I was just trying to keep everything clear . . ..*"

However, while spectacles may pose difficulties for physical activity, they were the preferred corrective option for 22% of a sample of adults participating in a range of non-competitive sporting activities that included 'athletics' [5].

Contact lenses are an alternative to spectacles and are worn by 6.6% of 15–64-year-olds across 11 European countries [6]. They are a popular solution for people engaging in sport, being the favoured vision correction option for 59% of a sample of adults engaging in non-competitive sporting activities [5]. However, whilst sport and physical activity are often cited as a reason for seeking contact lenses, not everyone can successfully wear contact lenses, with a mean dropout rate of 21.7% [7] driven primarily by discomfort.

The health benefits arising from regularly engaging in physical activity are widely recognised [8–10], with recreational running being a popular activity of choice. In Europe and Africa, running (9%) is the second highest recreational activity engaged in by adults after soccer (10%) [11]. Recreational running, by design, is accessible to all irrespective of age or level of fitness. The popularity of recreational running has increased over recent years, partly due to *parkrun*, the physical activity initiative taking place in public spaces across the United Kingdom (UK) and worldwide. In the UK in 2019, there were an average of 14.43±0.02 monthly parkrun finishers per 1000 persons, compared to 0.26±0.00 per 1000 persons in 2010 [12].

The number of people participating in recreational running, coupled with the high proportion of the population with refractive error, will require individuals to make a choice about their type of visual correction to use whilst running. In addition to the options of spectacles or contact lenses, it is possible that some people may choose to run uncorrected, inducing some form of visual blur. Support for the idea that people may choose to run uncorrected can be found though previous literature which has shown that mild levels of blur can be tolerated in dynamic sporting situations with little to no detriment in performance [13, 14].

Given the variety of refractive visual correction choices available and the potential disadvantages of spectacles for sport, we wanted to understand what correction choices are used by people for the popular participatory sport of recreational running. The aim of the current study was to identify the choices of correction (spectacles, contact lenses, no correction, or a combination of these options) made by those who participate in recreational running and require a distance vision correction, and to examine whether there are distinguishing demographics which are associated with particular choices of correction. It is expected that the results will support eyecare and sports practitioners by adding to the little evidence available regarding vision-related running advice.

## Method

### Questionnaire

A 26-item cross-sectional online questionnaire was developed with input from a group of vision professionals, academics and recreational runners requiring visual correction. The questionnaire was piloted with five recreational runners to ensure that wording was clear and concise. Following amendment, the questionnaire consisted of five sections and took approximately 20–30 minutes to complete:

Section 1: participant demographics including age and sex.

Section 2: visual information including selection of type of correction(s) used for everyday tasks from a provided list (termed 'general purpose wear'), selection of type of correction(s) used when running from a provided list, and how long visual correction had been needed. Participants were asked if they knew their refractive prescription, and if they did, the sphere (Dioptres; D), cylinder (D), axis (deg) and near addition (D) from their most recent prescription were captured for both right and left eyes.

Section 3: running habits including average distance of a run, average length of a run, weekly running frequency, how long the individual had been running, and whether participants ran in a habitual overground (outdoor) environment or on a treadmill (indoor).

Sections 4 and 5 explored satisfaction with visual corrections worn for running, and will be presented elsewhere.

### Participants

The study was approved by Anglia Ruskin University Research Ethics Committee and was conducted in accordance with the Declaration of Helsinki. Informed electronic written consent was obtained from all participants. Having been provided with information about the study, participants checked a box on the introductory page of the online survey to indicate their consent to take part prior to progressing to complete the survey.

Convenience sampling was used to recruit volunteer participants from the UK, Australia, Canada, and USA. No specific minimum sample size was sought for this exploratory study, but a sampling approach reflecting similar work in this area [5] was undertaken to maximise participation. A contact list of running clubs was collated from the website 'Time Outdoors' (timeoutdoors.com) and a total of 93 running clubs were emailed requesting that they share the survey link with their members, in addition to social media marketing. The survey was available on Jisc (Online surveys) from 1st October 2020 to 31st March 2021, and was closed after 2 weeks of null responses. Inclusion criteria required participants to be aged 18 years or older, consider themselves to be a recreational runner and to need refractive correction for everyday life. Participants were asked to declare that they met each of these criteria, and potential participants who did not meet the stated inclusion criteria were excluded from completion

of the questionnaire. Participants could withdraw from the study at any point by closing their browser window.

## Analysis

Refractive correction was classified on the basis of the Best Vision Sphere (BVS; Sphere + ½ cylinder) in the eye with the lower refractive power, representing the eye with better uncorrected vision.

Myopic prescriptions (negative BVS) were further categorised into low (up to -2.75DS), moderate (-3.00DS to <-6.00DS), and high (-6.00DS or greater) myopes [15]. Hyperopes (positive BVS) were not split into further categories as their uncorrected vision depends on the accommodation available, so their refractive error is less useful as a description of visual impairment without correction. Astigmatism was defined as a cylindrical component of at least 0.75DC present in at least one eye. Presbyopia was defined by the provision of a reading addition.

Statistical analyses were undertaken with IBM SPSS v28.0. Bivariate analyses were analysed as 2-tail independent t-tests for continuous variables, and chi-square analysis for categorical variables, with $p < 0.05$ considered significant. It was accepted that with a large number of comparisons, one in twenty will be significant by chance. A more stringent probability requirement was not adopted as the purpose was to identify appropriate parameters to enter into multiple regression analysis.

Strength of association for bivariate analyses is indicated by Cohen's d effect sizes for continuous variables and phi associations for categorical variables. Cohen's d values are interpreted as small (d = 0.2), medium (d = 0.5) or large (d = 0.8) [16]. Phi associations vary between 0 and 1, with larger values indicating closer association [17].

Binary logistic regression was used to determine which of the variables identified in bivariate analyses were independently associated with including or avoiding types of correction for running. A backwards stepwise (likelihood ratio) method was used for this exploratory analysis. Any tolerance values <0.1 or variance inflation factors (VIF) >10 would be flagged for further investigation with regard to multicollinearity [17].

## Results

The questionnaire was started by 957 people and was fully answered by 941 self-declared recreational runners (S1 File; also available at https://doi.org/10.25411/aru.25109270.v1). Of those who completed the questionnaire, 525 (55.8%) were female, 413 (43.9%) were male, and 3 (0.3%) preferred not to provide their sex. Mean age was 47 years (SD 12, range 18–79). Participants reported being a runner for an average of 13 years (SD 12, range <1–65). The mean number of runs per week reported was 3.8 (SD 1.5, range 1–14), with an average time spent running of 3.6 hours/week (SD 2.4, range 0.3–35), and an average distance run of 33.4 km/week (SD 20.3, range 3–161). The mean duration that visual correction had been required for everyday use was 26 years (SD 14, range 0.3–65). For general purpose wear, 930 people (98.9%) had spectacles, of whom 378 (40.2%) had multifocal spectacles (bifocal or progressive power (varifocal) lenses). Contact lenses were worn for general purposes by 395 people (42.0%).

A subset of 423 participants provided their refractive correction. Of the subset, 116 (27.4%) were hyperopic, and 307 (72.6%) were myopic, with 159 low myopes, 104 moderate myopes, and 44 high myopes. Astigmatism was present in 78 (18.4%) people, and 70 (16.4%) provided a presbyopic reading addition. The mean BVS in the eye with the lower refractive power was -1.64D (SD 3.64, range -12.50 to +10.75).

**Table 1. Cross-tabulation of visual correction(s) worn by participants for general purposes in everyday life compared to when participating in running.**

| Visual correction type(s) worn | | Running | | | | | | | |
|---|---|---|---|---|---|---|---|---|---|
| | | CL only | Spectacles & CL | Spectacles only | None | Spectacles & none | CL & none | Spectacles, CL or none | Total |
| General purpose wear | CL only | 9 | 0 | 0 | 2 | 0 | 0 | 0 | 11 (1.2%) |
| | Spectacles & CL | 127 | 165 | 21 | 12 | 2 | 19 | 14 | 360 (38.3%) |
| | Spectacles only | 10 | 15 | 337 | 114 | 28 | 1 | 3 | 508 (54.0) |
| | None | 0 | 0 | 0 | 0 | 0 | 0 | 0 | 0 (0%) |
| | Spectacles & none | 0 | 0 | 5 | 15 | 17 | 0 | 1 | 38 (4.0%) |
| | CL & none | 0 | 0 | 0 | 0 | 0 | 0 | 0 | 0 (0%) |
| | Spectacles, CL or none | 1 | 2 | 2 | 3 | 2 | 9 | 5 | 24 (2.6%) |
| | Total | 147 (15.6%) | 182 (19.3%) | 365 (38.8%) | 146 (15.5%) | 49 (5.2%) | 29 (3.1%) | 23 (2.4%) | 941 |

CL = contact lenses. None = no visual correction worn.

## Prevalence of visual correction use for general purposes and running

Table 1 shows the types of visual correction worn by participants for general tasks in their day to day lives (general purpose wear) compared to the visual correction(s) worn when participating specifically in running. Participants could select options of spectacles, contact lenses (CLs) and going without correction (none), along with combinations of these options. Table 1 also shows that all participants used a visual correction at least some of the time for general purposes (none = 0%), and that very few people were wholly reliant on contact lenses (CL only = 1.2%). When running, the commonest visual correction solution was always to run in spectacles (38.8%), while 15.5% always ran without correction and 15.6% always ran with contact lenses. In terms of common specific behaviour changes, 127 people (13.5%) who used both spectacles and contact lenses for general purposes only used contact lenses when running, and 114 people (12.1%) who only wore spectacles for general purposes only ran without correction.

Table 2 presents the data from Table 1 in an alternative format to identify the proportions of people including and avoiding different visual corrections for general purposes and for running. While almost all people (98.9%) include spectacles in their visual correction for general purposes, this falls to 65.8% who will wear spectacles for running. Similar numbers of those who have access to contact lenses for general purposes (42.0%) use them at least some of the time for running (40.5%). While very few people (6.6%) go without a correction for general purposes at least sometimes, this rises to 26.2% who will run without correction.

**Table 2. Visual correction types worn and avoided for general purposes and when running.**

| Visual correction type(s) worn | General purpose wear | | Running | |
|---|---|---|---|---|
| | Number | % | Number | % |
| Includes no correction | 62 | 6.6 | 247 | 26.2 |
| Includes CL | 395 | 42.0 | 381 | 40.5 |
| Includes spectacles | 930 | 98.9 | 619 | 65.8 |
| Avoids no correction | 879 | 93.4 | 694 | 73.8 |
| Avoids CL | 546 | 58.0 | 560 | 59.5 |
| Avoids spectacles | 11 | 1.1 | 322 | 34.2 |

## Demographic differences between people who include and avoid different visual corrections for running

We next explore the differences between people who include and avoid the 3 types of visual correction for running (no correction, spectacles, contact lenses). Bivariate analyses were used to identify parameters where there were significant differences between those who included and avoided a visual correction type. Predictor variables assessed were sex, age, running duration, distance run, time run, runs per week, correction duration, availability of contact lenses for general wear, and use of multifocal spectacles for general wear.

Where indicated, parameters with significant relationships were then included in logistic regression analyses to determine the parameters explaining most of the independent variance in the data.

**No correction.**   People who would run without correction (n = 247) have worn correction for less time (20.2 (SD 14.4) years) than those who would not (n = 694; 28.0 (SD 13.8) years) (t (930) = -7.5, p<0.001, d 0.56) with a medium effect size. People with contact lenses for general purposes were less likely to run without correction (chisquare 28.7 (df1), p < .001, phi 0.18). No further parameters were significant.

Binary logistic regression (Table 3) indicated that both these parameters were significant predictors of running without visual correction. The likelihood of running without correction decreases if contact lenses are available (odds ratio 0.43) and for every year that visual correction has been worn (odds ratio 0.96).

For the subset of 423 people who provided their refractive correction, Fig 1 shows that those who were prepared to run without correction had lower refractive errors (i.e., BVS values closer to zero). This is supported by an independent t-test indicating that those who include no correction (n = 87) had lower refractive errors (mean -0.17 (SD 1.96) D) than those who avoid no correction (n = 336; mean -2.02 (SD 3.87) D) (t(421) 4.31, p<0.001, d 0.52) with a medium effect size.

**Spectacles.**   People who wear spectacles for running (n = 619) were older (48.1 (SD 12.2) years) than those who did not (n = 321; 45.6 (SD 12.0) years) (t(938) = 3.0, p = 0.003, d 0.21) with a small effect size. They had also worn correction for longer (27.8 (SD 14.2) years) than those who avoided spectacles (22.4 (SD 14.1) years) (t(930) = 5.4, p<0.001, d 0.38) with a small effect size. Sex was significant (chisquare 15.3 (df1), p<0.001, phi 0.13), with women more likely to avoid spectacles for running. People who did not have contact lenses available for general use were more likely to wear spectacles for running (chisquare 42.5 (df1), p<0.001, phi 0.21), and people using multifocal spectacles for general use were slightly more likely to use spectacles for running (chisquare 5.9 (df1), p = 0.015, phi 0.08).

The five significant parameters were entered into binary logistic regression (Table 4) which indicated that correction duration and availability of contact lenses were significant predictors

**Table 3.  Binary logistic regression of the likelihood of including no correction in the visual correction(s) worn for running.**

| Included parameters | B (SE) | Odds ratio (OR) | 95% CI for OR |
|---|---|---|---|
| Constant | 0.26 (0.16) | | |
| CL available for general purposes | -0.85 (0.17)* | 0.43 | 0.31–0.59 |
| Correction duration (years) | -0.04 (0.006)* | 0.96 | 0.95–0.97 |

Nagelkerke $R^2$ = 0.13, Model chi-square (df 2) = 83.96, p<0.001. *p<0.001. Parameters entered into the model were the availability of contact lenses for general purpose wear (no/yes) and the duration that correction had been worn for general purposes (years).

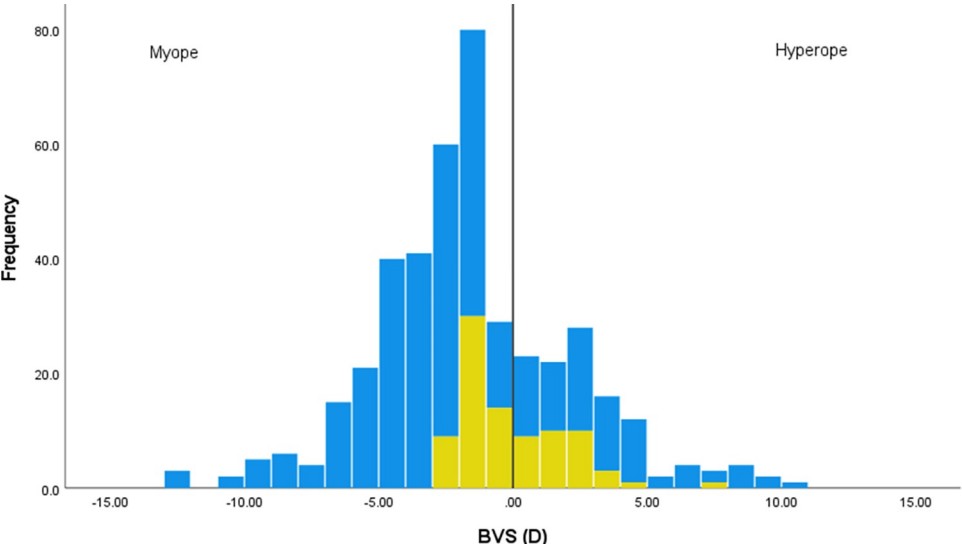

**Fig 1. Histogram representing the refractive error of those who would run without correction (yellow) or would avoid running without correction (blue).** Refractive error is given by the best vision sphere (BVS) of the eye with the lower refractive power for the 423 people providing their refractive correction. The vertical line represents a BVS of zero, with myopes (negative BVS) to the left of the line, and hyperopes (positive BVS) to the right.

of running in spectacles. The likelihood of running in spectacles decreases significantly if contact lenses are available (odds ratio 0.40) and increases for every year that visual correction has been worn (odds ratio 1.03).

For the subset of 423 people who provided their refractive correction, Fig 2 shows that spectacle use and avoidance for running is spread across the range of refractive errors. Those who include spectacles (n = 273; mean -1.73 (SD 3.77) D) have statistically similar refractive errors to those who avoid spectacles (n = 150; mean -1.48 (SD 3.40) D) (t(421) -0.68, p = 0.50).

**Contact lenses.** People who wear contact lenses for running (n = 381) were younger (44.1 (SD 11.1) years) than those who did not (n = 559; 49.4 (SD 12.4) years) (t(938) = 6.7, p<0.001, d 0.45) with a small effect size. Sex was significant (chisquare 26.3 (df1), p<0.001, phi 0.17), with women more likely to use contact lenses for running. Most people with contact lenses available for general wear used them for running (chisquare 661, df1, p<0.001, phi 0.84), and multifocal spectacle wearers were less likely to use contact lenses for running (chisquare 47.8, df1, p<0.001, phi 0.23).

**Table 4. Binary logistic regression of the likelihood of including spectacles in the visual correction(s) worn for running.**

| Included parameters | B (SE) | Odds ratio (OR) | 95% CI for OR |
|---|---|---|---|
| Constant | 0.53 (0.19) | | |
| CL available for general purposes | -0.91 (0.15)* | 0.40 | 0.30–0.54 |
| Correction duration (years) | 0.03 (0.005)* | 1.03 | 1.02–1.04 |
| Sex | -0.27 (0.15)** | 0.77 | 0.57–1.03 |

Nagelkerke $R^2$ = 0.11, Model chi-square (df 2) = 76.83, p<0.001. *p<0.001. **not significant (p = 0.08). Parameters entered into the model were age (years), sex (male/female), use of multifocal spectacles for general purpose wear (no/yes), the availability of contact lenses for general purpose wear (no/yes) and the duration that correction had been worn for general purposes (years).

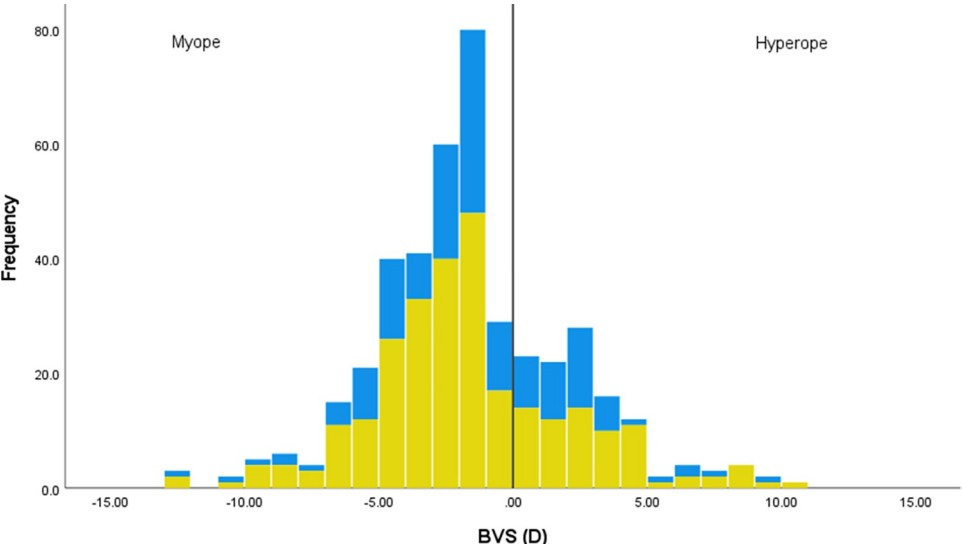

**Fig 2. Histogram representing the refractive error of those who would run with spectacles (yellow) or would avoid running with spectacles (blue).** Refractive error is given by the best vision sphere (BVS) of the eye with the lower refractive power for the 423 people providing their refractive correction. The vertical line represents a BVS of zero, with myopes (negative BVS) to the left of the line, and hyperopes (positive BVS) to the right.

The four significant parameters were entered into binary logistic regression (Table 5) which indicated that the availability of contact lenses was the sole significant predictor of running in contact lenses (odds ratio 138).

For the subset of 423 people who provided their refractive correction, Fig 3 shows that myopes are particularly frequent users of contact lenses. This is supported by an independent t-test indicating that the refractive error of contact lens users (n = 251; mean -2.32 (SD 3.46) D) is more negative than that of non-contact lens users (n = 172; mean -0.66 (SD 3.67) D) (t (421) -4.74, p<0.001, d -0.47) with a small effect size.

## Discussion

This study used a self-report questionnaire to understand, from those who participate in recreational running and require a distance vision correction, what their choices of visual correction are and whether there are distinguishing demographics associated with particular choices of correction. A strength of the study is the large number (941) of participants sampled, reflecting a broad spectrum of recreational runners whose running demographics represent an 'above active' group compared to recommendations [9]. A further strength of the study is its novelty, with very few studies existing with which the present results can be compared. A

**Table 5. Binary logistic regression of the likelihood of including contact lenses in the visual correction(s) worn for running.**

| Included parameters | B (SE) | Odds ratio (OR) | 95% CI for OR |
|---|---|---|---|
| Constant | -2.84 (0.19) | | |
| CL available for general purposes | 4.97 (0.25)* | 143.46 | 88.09–233.66 |

Nagelkerke $R^2$ = 0.75, Model chi-square (df 1) = 767.65, p<0.001. *p<0.001. Parameters entered into the model were age (years), sex (male/female), use of multifocal spectacles for general purpose wear (no/yes), and the availability of contact lenses for general purpose wear (no/yes).

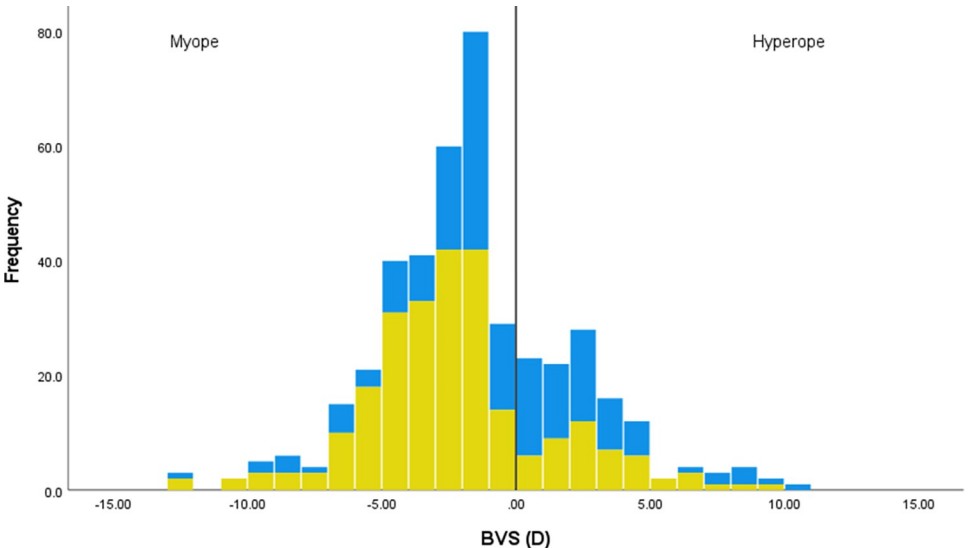

**Fig 3. Histogram representing the refractive error of those who would run with contact lenses (yellow) or would avoid running with contact lenses (blue).** Refractive error is given by the best vision sphere (BVS) of the eye with the lower refractive power for the 423 people providing their refractive correction. The vertical line represents a BVS of zero, with myopes (negative BVS) to the left of the line, and hyperopes (positive BVS) to the right.

limitation of the study is that the volunteers choosing to take part in the study were self-selected, and thus may not fully represent all recreational runners who need to wear visual correction.

Visual correction choices did change for running as compared to those worn for general purposes. Fewer people wore spectacles for running (65.8%) than for general purposes (98.9%) (Table 2), although solely using spectacles for running (38.8%) was the most common specific correction choice (Table 1). As alternatives to spectacles, those who have contact lenses do wear them for running (Tables 2 & 5), and a greater proportion of people will go without correction for running than for general purposes (26.2% compared to 6.6%; Table 2).

However, there is no evidence that running behaviour is associated with choice of visual correction. None of the running demographics considered (running duration, distance run, time run, runs per week) were significantly associated with the likelihood of including or avoiding the three types of vision correction (spectacles, contact lenses, no correction). This suggests that for those who require a visual correction, or run in a certain type of correction, the visual correction does not present a barrier to engaging in recreational running.

Running without visual correction is a common visual choice, with 15.5% of the sample only running without correction (Table 1), and 26.2% running at least sometimes without their correction (Table 2). People were more likely to run without correction if they did not have contact lenses available to them, had been wearing visual correction for a shorter time (Table 3), or had lower refractive errors (Fig 1).

For myopes, the highest refractive error where the inclusion of no correction was reported was -3.00D (Fig 1). Other than this one individual, all other corrections were ≤-2.50D. The vision associated with a myopic refractive error of 2.50–3.00D would be expected to be in the region of 0.86–1.0logMAR [18] but with considerable individual variation due to factors such as pupil size or interpretation of blur [19]. The range of hyperopic corrections of those who were prepared to run without correction was wider and included values up to a maximum of +7.50D. It is difficult to predict the uncorrected vision of people with hyperopic corrections as

younger individuals can use accommodation to overcome the refractive error, particularly at lower values.

These results support previous research which has demonstrated how mild levels of blur can be tolerated with little to no detriment in sporting performance [13, 14]. What level of blur can be tolerated without detriment to recreational running performance is not known, but these results suggest that at least some individuals are prepared to run with vision as poor as 1.0logMAR (6/60 or 20/200 Snellen). Of note, best corrected visual acuity of 6/60 is the boundary for sight impaired registration in the UK [20] and also for entry into many vision impaired sports [21].

Although spectacles were the most common correction type for running, a third of the sample (33.1%) actively avoid running in spectacles despite wearing them for general purposes. This finding highlights that spectacles have limitations for a subset of our sample when running as compared to general purpose use. Future analysis of sections 4 & 5 of this questionnaire will consider reasons why people avoid running whilst wearing spectacles.

The high frequency of spectacle use for running found in this study contrasts with previous work [5] that reported contact lenses being the favoured corrective option (58.8%) for non-competitive sporting activity, with far fewer (22.0%) choosing spectacles. 'No correction' was not an included option in that study. The differences in findings are likely to reflect differences in the types of activities sampled, with most of the sporting activities reported in the previous study [5] being dynamic, high-intermittent sub-maximal intensity and often contact sports. 'Running' was not a specific category although it was captured within 'athletics'. In comparison, recreational running is a lower intensity non-contact sport. Collectively, these results suggest that recreational running is an accessible activity for those who require a visual correction and can only wear spectacles i.e., are unable to wear contact lenses.

In the present analysis, spectacles were less likely to be worn for running if contact lenses were available, and more likely to be worn if visual correction had been worn for longer. Longer correction duration has been previously associated with wearing spectacles for sport [5]. It is not clear why having worn a visual correction for longer might be associated with increased likelihood of wearing spectacles and also of avoiding no correction for running. It is possible that people who have worn spectacles for longer may be more accepting of this visual correction and feel less need to seek alternative solutions.

Contact lenses are a successful solution as a visual correction for running for those who wear them. The vast majority of those who have contact lenses (42.0%) use them at least sometimes for running (40.5%; Table 2), with 15.6% of the sample using contact lenses as their sole visual correction for running compared to 1.2% solely using contact lenses for general purposes (Table 1). Having access to contact lenses was significantly associated with a decreased likelihood of running in spectacles (Table 4) or without correction (Table 3).

The data suggest that higher myopes were more likely to use contact lenses (Fig 3), and these runners would not have the option of running without correction due to a higher level of visual impairment. Bivariate analyses also suggested that women and younger people were more likely to use contact lenses, in keeping with other studies [22]. Paying particular attention to signposting male and older runners towards contact lenses may be useful, as these runners may be less likely to have considered this visual option.

## Recommendations for eye care practitioners and sports practitioners

We have investigated the choices of visual refractive correction made by those who participate in recreational running and require a distance vision correction. Key findings are that recreational runners are prepared to run in any form of visual correction. Spectacles are the most

common solution for running, but a substantive portion choose to avoid wearing them to run. People are motivated out of spectacles if they have access to contact lenses, but if they do not have contact lenses and have a low refractive error, many run without correction. Further work is needed to establish any impact of running without correction on safety and performance.

Eye care practitioners and sports practitioners [23] are encouraged to explore the range of visual correction options with people who run, and to signpost them to contact lenses as an option if this has not previously been considered. However, visual correction choice does not create a barrier to recreational running and can be successfully participated in with any form of vision correction.

## Supporting information

**S1 File. Complete raw data.** Data of participant demographics, visual correction choices, and running habits.
(XLSX)

## Author Contributions

**Conceptualization:** Keziah Latham, Georgia Dunn, Andrew Morrison, Matthew A. Timmis.

**Data curation:** Georgia Dunn.

**Formal analysis:** Keziah Latham, Georgia Dunn, Matthew A. Timmis.

**Investigation:** Georgia Dunn.

**Methodology:** Keziah Latham, Georgia Dunn, Andrew Morrison, Matthew A. Timmis.

**Project administration:** Georgia Dunn.

**Supervision:** Keziah Latham, Andrew Morrison, Matthew A. Timmis.

**Visualization:** Keziah Latham.

**Writing – original draft:** Keziah Latham, Matthew A. Timmis.

**Writing – review & editing:** Keziah Latham, Georgia Dunn, Andrew Morrison, Matthew A. Timmis.

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
