## [Decision Letter · Decision Letter 0]

15 Jan 2024

PONE-D-23-34308What choices of visual correction do people make for recreational running?PLOS ONE

Dear Dr. Latham,

Thank you for submitting your manuscript to PLOS ONE. After careful consideration, we feel that it has merit but does not fully meet PLOS ONE’s publication criteria as it currently stands. Therefore, we invite you to submit a revised version of the manuscript that addresses the points raised during the review process.

We look forward to receiving your revised manuscript.

Kind regards,

Ugochukwu Anthony Eze

Academic Editor

PLOS ONE

Reviewers' comments:

Reviewer's Responses to Questions

**Comments to the Author**

1. Is the manuscript technically sound, and do the data support the conclusions?

Reviewer #1: Yes

Reviewer #2: No

2. Has the statistical analysis been performed appropriately and rigorously? 

Reviewer #1: Yes

Reviewer #2: Yes

3. Have the authors made all data underlying the findings in their manuscript fully available?

Reviewer #1: Yes

Reviewer #2: No

4. Is the manuscript presented in an intelligible fashion and written in standard English?

Reviewer #1: Yes

Reviewer #2: No

5. Review Comments to the Author

Reviewer #1: The is a valuable and interesting study that will be useful in the eye care of runners. The study employed robust statistical methods to achieve its aim of highlighting the choices of visual correction by recreational runners and identifying characteristics associated with them. Though majority use spectacles for general activities, a considerable proportion do not do so when running, unlike those who use contact lenses with nearly equal proportions for both activities. This suggests that there may be problems with use of spectacles for running which the authors rightly identified as an area for future research. Females and younger runners tended to use contact lenses more.

Few minor revisions could help enrich and improve clarity of the manuscript as follows:

1. In line with scientific writing standards, Line 25 could be rephrased to avoid starting the sentence with a number

2. the results are well presented and mostly clear. A few areas would require clearer presentation. It is suggested that line 198 to 200 be rephrased to convey the finding that spectacles are not worn for running by a proportion of respondents; as presented, it appears as though an inference is being drawn in the results section (which should ideally be done under discussion instead).

3. The discussion section of the manuscript is adequately laid out with interpretation of and elaboration of implications of the study findings as well as relation with existing studies. This section could be made more concise by minimising repetitions. For example, line 320 to 322 which have been expressed in the introduction and methods section and could be deleted. Also lines 324 to 325 appear to be repetition of results that had been stated in the results section. Furthermore lines 332 to 335 contains the same information as line 366 to 369, these could be merged.

The manuscript provides a very useful contribution to knowledge about visual correction preferences of recreational runners with visual impairment that would help advise their care. It is recommended that the manuscript be accepted for publication.

Reviewer #2: Abstract

The summary of the method used in the study was not clearly stated in the abstract.

The abstract lacked numbers, percentage values as well as statistical analysis findings that will help for the easy understanding of the study findings.

Introduction

The introduction lacked relevant information including justification for the study.

Method

The method lacked details.

The authors spent so much time explaining the questionnaire rather than the method.

The study design, population, sample size, and exclusion criteria employed in the study were not clear.

How was the questionnaire developed? Was it validated and how was it validated?

How many participants were recruited and how many ended up participating in the study?

Method used in recruiting participants was not detailed.

Was consent obtained from participant prior to data collection and what type of consent was obtained prior to data collection?

Results

The results were confusing and not logically presented.

Most of the information in the tables are redundant and not clear.

I expected the authors to have a table with the characteristics of the study participants.

I struggled to understand the information in table 1 and 2 as well as their purpose and relevance.

The meaning of general purpose in table 2 is not clarified.

The interpretation of the results was simply repetition of what was on the table.

The potential source of bias including confounding variables and how they were adjusted was not clearly stated.

Discussions

There was basically no discussion including comparison of the research findings with previous studies.

There was no need to repeat the results in the discussion.

The manuscript had no strengths and limitations.

Conclusion

The conclusion does not align with the study objectives.

6. PLOS authors have the option to publish the peer review history of their article (what does this mean?). If published, this will include your full peer review and any attached files.

Reviewer #1: No

Reviewer #2: No

---

## [Author Response · Author response to Decision Letter 0]

6 Feb 2024

Please see the Cover Letter enclosed as an uploaded document for a point by point consideration of the reviewers' comments.

---

## [Decision Letter · Decision Letter 1]

24 May 2024

What choices of visual correction do people make for recreational running?

PONE-D-23-34308R1

Dear Dr. Latham,

We’re pleased to inform you that your manuscript has been judged scientifically suitable for publication and will be formally accepted for publication once it meets all outstanding technical requirements.

Kind regards,

James Mockridge

Staff Editor

PLOS ONE

Additional Editor Comments (optional):

Reviewers' comments:

Reviewer's Responses to Questions

**Comments to the Author**

1. If the authors have adequately addressed your comments raised in a previous round of review and you feel that this manuscript is now acceptable for publication, you may indicate that here to bypass the “Comments to the Author” section, enter your conflict of interest statement in the “Confidential to Editor” section, and submit your "Accept" recommendation.

Reviewer #1: All comments have been addressed

Reviewer #2: All comments have been addressed

2. Is the manuscript technically sound, and do the data support the conclusions?

Reviewer #1: Partly

Reviewer #2: Yes

3. Has the statistical analysis been performed appropriately and rigorously? 

Reviewer #1: Yes

Reviewer #2: Yes

4. Have the authors made all data underlying the findings in their manuscript fully available?

Reviewer #1: Yes

Reviewer #2: Yes

5. Is the manuscript presented in an intelligible fashion and written in standard English?

Reviewer #1: Yes

Reviewer #2: Yes

6. Review Comments to the Author

Reviewer #1: (No Response)

Reviewer #2: I don't have any more comments. All my comments have been addressed. .

7. PLOS authors have the option to publish the peer review history of their article (what does this mean?). If published, this will include your full peer review and any attached files.

Reviewer #1: No

Reviewer #2: No

---

## [Editor Report · Acceptance letter]

13 Jun 2024

PONE-D-23-34308R1 

PLOS ONE

Dear Dr. Latham, 

I'm pleased to inform you that your manuscript has been deemed suitable for publication in PLOS ONE. Congratulations! Your manuscript is now being handed over to our production team.

Kind regards, 

on behalf of

Dr James Mockridge 

Staff Editor

PLOS ONE